# Aging and Metabolic Reprogramming of Adipose-Derived Stem Cells Affect Molecular Mechanisms Related to Cardiovascular Diseases

**DOI:** 10.3390/cells12242785

**Published:** 2023-12-07

**Authors:** Paul Holvoet

**Affiliations:** Division of Experimental Cardiology, Katholieke Universiteit Leuven, 3000 Leuven, Belgium; paul.holvoet@kuleuven.be

**Keywords:** adipose, stem cells, oxidative stress, inflammation, atherosclerosis, heart disease

## Abstract

We performed a systematic search of the PubMed database for English-language articles related to the function of adipose-derived stem cells in the pathogenesis of cardiovascular diseases. In preclinical models, adipose-derived stem cells protected arteries and the heart from oxidative stress and inflammation and preserved angiogenesis. However, clinical trials did not reiterate successful treatments with these cells in preclinical models. The low success in patients may be due to aging and metabolic reprogramming associated with the loss of proliferation capacity and increased senescence of stem cells, loss of mitochondrial function, increased oxidative stress and inflammation, and adipogenesis with increased lipid deposition associated with the low potential to induce endothelial cell function and angiogenesis, cardiomyocyte survival, and restore heart function. Then, we identify noncoding RNAs that may be mechanistically related to these dysfunctions of human adipose-derived stem cells. In particular, a decrease in let-7, miR-17-92, miR-21, miR-145, and miR-221 led to the loss of their function with obesity, type 2 diabetes, oxidative stress, and inflammation. An increase in miR-34a, miR-486-5p, and mir-24-3p contributed to the loss of function, with a noteworthy increase in miR-34a with age. In contrast, miR-146a and miR-210 may protect stem cells. However, a systematic analysis of other noncoding RNAs in human adipose-derived stem cells is warranted. Overall, this review gives insight into modes to improve the functionality of human adipose-derived stem cells.

## 1. Introduction

Adipose-derived stem cells (ASCs) are easily acquired with high yields and, therefore, are an ideal stem cell source [1]. The International Fat Applied Technology Society, renamed the International Federation for Adipose Therapeutics and Science, reached a consensus referring to adipose-derived stem cells for all plastic-adherent, multipotent cell populations isolated from adipose tissue, instead of referring to adipose stromal cells, adipose-derived adult stem cells, or adipose mesenchymal stem cells [2]. ASCs are commonly isolated from the stromal vascular fraction (SVF) of adipose tissue. ASCs in white adipose tissue (WAT) have the potential to become preadipocytes, subsequently differentiating into mature adipocytes via adipogenesis involving the activation of peroxisome proliferator-activated receptor gamma (PPARγ). Anatomically separated WAT depots, namely subcutaneous WAT (S-WAT) and visceral WAT (V-WAT), are known to be functionally distinct. S-WAT expands to store excess lipids, thus preventing ectopic lipid disposition and organ damage, while the main function of V-WAT is to cushion and protect the visceral organs [3]. However, apart from adipogenesis, ASCs can acquire properties of specialized cells or induce the differentiation of other cell types, among them endothelial cells (ECs) [4], vascular smooth muscle cells [5], and cardiomyocytes [6]. In culture, ASCs retain markers in common with other mesenchymal stromal/stem cells, including CD90, CD73, CD105, and CD44, and remain negative for CD45 and CD31 [7,8,9,10].In addition, the immunological reactivity of ASCs is low because of the low expression of immunogenic surface antigens (CD40, CD40L, CD80, and CD86) and major histocompatibility complex II, allowing allogeneic use [11]. Finally, ASCs exert paracrine function by producing cytokines and growth factors, such as vascular endothelial growth factor (VEGF), fibroblast growth factor (FGF), and insulin-like growth factor 1 (IGF-1) [12,13].

This review specifically aims at understanding mechanisms by which ASCs may inhibit atherosclerosis and improve heart function and thus protect against cardiovascular diseases. The initial step in atherosclerosis is endothelial dysfunction through mechanical shear stress and chemical stress induced by high glucose, high LDL and low HDL, high levels of reactive oxygen species (ROS) and oxidized LDL, and ANG-II. Injured endothelium attracts inflammatory cells, among them macrophages. Stress induces the polarization of macrophages from an anti-inflammatory M2 to an inflammatory M1 phenotype. M1 macrophages accumulate lipids and differentiate into foam cells. ROS and lipids induce cell death, thereby destabilizing atherosclerotic plaques. High oxidative stress also increases Th1/Th17 and decreases Th2/Treg cell immune response, further enhancing inflammation and cell death [14,15]. 

We particularly reviewed the role of ASCs exerting therapeutic effects that rely on paracrine secretion. Indeed, ASCs secrete several cytokines, growth factors, and chemokines that modulate oxidative stress, inflammation, immune responses, angiogenesis, and apoptosis in damaged vascular and heart cells [16]. In the heart, the pericardium consists of a thin fibrous layer, nerves, a vascular network, and adipose mass. It also comprises mesenchymal cells, so-called formed cardiac colony-forming unit fibroblasts that give rise to all mesodermal lineages, including smooth muscle, bone, cartilage, adipose, endothelial, and heart muscle cells [17]. Pericardial ASCs constituted intrinsic properties toward myogenesis and vessel formation and thus provided more potent structural repair, translating into functional amelioration after myocardial injury [18].

We discussed first the effect of ASCs in preclinical models, giving insight into underlying mechanisms. Then, we reviewed the clinical data. Because clinical trials did not reproduce successful treatments with ASCs in preclinical models, we tried to identify molecular explanations for the loss of function of ASCs, focusing on aging and metabolic reprogramming. We considered changes in the expression of noncoding RNAs in addition to changes in protein or lipid content because of their association with metabolic and cardiovascular diseases [19].

Currently, one aims to move from cell-based to cell-free therapy with extracellular exosomes of stem cell origin [20], further reducing the immunogenic response and thus improving the safety and increasing the stability upon storage and obtaining a better targeting of the cells and tissues to be treated [21]. Previously, we showed the role of exosome-mediated cell-to-cell communication in affecting pathways involved in the pathogenesis of cardiovascular diseases [22,23]. Herein, we discussed particularly the role of exosomes secreted by ASCs. 

## 2. Information Resources, Search, and Study Selection

Figure 1 illustrates the strategy and outcome of the literature search.

We searched the PubMed database for English-language articles related to adipose-derived stem cells in mechanisms related to cardiovascular diseases. The search strategy encompassed a MESH search: *(*‘Angiogenesis’ [Mesh] OR ‘Atherosclerosis’ [Mesh] *OR* ‘Myocardial Infarction’ [Mesh] OR ‘Cardiomyopathy’ [Mesh]) OR ‘Heart’ [Mesh]) AND (‘Adipose-derived stem cells’ [Mesh]), identifying 1327 titles. The figure illustrates reasons for the exclusion of papers from this review paper: 552 papers reported only on in vitro experiments on the differentiation of stem cells including incubation with other cells or other components, such as growth factors and their interaction with scaffolds; 82 papers reported on in vivo experiments, for example, comparing fresh, uncultured cells with cells cultured under different conditions but without any mechanistic explanation for diverging results; 388 papers dealing with, in particular, angiogenesis in other diseases like wound healing, bot formation, and oncogenesis and referring to possible value for cardiovascular diseases; 15 papers published by the same authors in several journals but with basically the same content; 129 not reporting original experimental data like review papers, comments, and editorials. In addition, twenty-five papers were excluded because of flawed designs (too small a number of biological replicates or flawed statistics, for example, by not using consistent numbers of biological replicates) and thirty-six because full text was not available in English. The additional thirty-seven references give background information about the properties of ASCs and mechanisms in angiogenesis, atherosclerosis, and ischemia.

## 3. Adipose-Derived Stem Cells Preserve EC Function in Preclinical Models

Endothelial dysfunction links metabolic abnormalities, such as obesity, type 2 diabetes, and dyslipidemia, to cardiovascular diseases [24]. ASCs may protect ECs and their functions by secreting paracrine factors rather than differentiating into mature ECs [25]. Exosomes from ASCs inhibited the expression of miR-342-5p in ECs, thereby reverting their apoptosis [26] (Figure 2).

In a hindlimb ischemic mouse model, ASCs mediated angiogenesis by inducing the *FMS-related receptor tyrosine kinase 1* (*FLT-1*) gene encoding VEGFR1, the *kinase insert domain receptor* (*FLK-1*) gene encoding VEGFR2, and the *angiopoietin two* (*ANG-II*) gene. ASCs also increased VE-Cadherin, hepatic growth factor (HGF), CD31, myogenic factor (MYF)-5, and TGF-β1 [27]. In addition, hypoxic human ACSs, but not normoxic ASCs, secreted leptin that promoted EC angiogenesis through HIF-2α but not HIF-1α [28]. Furthermore, hypoxia-stimulated ECs secreted TGF-β1, which promoted the viability of ASCs that, in turn, enhanced the angiogenesis ability of human microvascular ECs. However, miR-20a suppressed TGF-β1 [29]. Exosomes secreted by ASCs overexpressing miR-21 promoted vascularization involving HIF-1α, VEGF, stromal cell-derived factor 1 (SDF-1), phosphorylated AKT, and phosphorylated mitogen-activated protein kinase (ERK1/2) [30]. In addition, exosomal miR-181b induced HIF-1α and VEGF and downregulated the tissue inhibitor of MMP-3, thereby increasing migration ability [31].

Human ASCs overexpressing the high mobility group box 1 (HMGB1) protein increased post-ischemic angiogenesis more than control ASCs through increasing VEGF activity [32]. Engrafted mouse ASCs also induced angiogenesis by activating the mechanistic target of the rapamycin kinase (mTOR) pathway, associated with reduced inflammatory neutrophil/macrophage infiltration, the secretion of pro-inflammatory IL-1β and TNF-α, and apoptosis [33]. 

Exosomes from ACSs overexpressing glyoxalase-1 (GLO-1) protected ECs from high-glucose stress. They preserved the heart muscle structure and angiogenesis by inducing nitric oxide synthase (NOS), AKT, and MAPK (ERK) and inhibiting the *Jun proto-oncogene* expression, ROS release, and the activation of the inflammatory nucleotide-binding and oligomerization domain-like receptor family pyrin domain-containing 3 (NLRP3). They also inhibited caspase-1 and IL-1β pathways, protecting ECs and cardiomyocytes against cell death [34]. Netrin-1 overexpression increased the vasculogenic capacity of ASCs by inducing metalloproteinase (MMP)-2 and MMP-9, VEGF, and placental growth factor (PlGF) [35] (Figure 2). 

## 4. Adipose-Derived Stem Cells Prevent Atherosclerosis in Preclinical Models

Autologous ASCs reduced oxidative stress by increasing the anti-oxidative NAD(P)H quinone oxidoreductase 1 (NQO1) and heme oxygenase-1 (HMOX1 or HO-1), thereby preventing macrophage polarization from M2 to M1 [36]. In addition, exosomal miR-148a-3p attenuated M1 macrophage polarization and inflammation by activating *Notch* signaling [37], miR-301a by inhibiting toll-like receptor (TLR)-4 [38], and miR-451 by inhibiting the migration inhibitory factor (MIF) [39]. In addition, blocking miR-130b-3p with H19 in ASCs induced PPARγ and STAT3, promoting macrophage M2 polarization [40]. Hypoxia-induced circular RNA derived from small nucleolar RNA host gene 11 (Circ-SNHG11) in exosomes from ASCs suppressed the high glucose-induced EC damage and retained M2-like macrophage polarization by inhibiting miR-144-3p [41] (Figure 3).

Autologous ASCs reduced inflammation by decreasing the release of vascular cell adhesion molecule (VCAM)-1, intercellular adhesion molecule (ICAM)-1, tumor necrosis factor-α (TNF-α), and nuclear factor-κB (NF-κB ) in a rat ischemia-reperfusion model [36]. In addition, ASCs inhibited atherosclerosis in a high-fat diet rabbit model by inhibiting the accumulation of oxidized LDL through increased CD36 protein, reverting M1 macrophage polarization, IL-6, and TNF-α release, and apoptosis [42]. Exosomes from ASCs enriched in stanniocalcin-1 (STC-1), a secreted glycoprotein that protects against inflammation, apoptosis, and necrosis, inhibited the production of NLRP3, caspase-1, and IL-1β in ECs, thereby improving angiogenesis [43]. MiR-145 and miR-221 were enriched in ASC exosomes and downregulated pro-inflammatory IL-6, NF-κB, and TNF-α while upregulating anti-inflammatory IL-10 [44]. Exosomal miR-17-5p blocked NLRP3 signaling in ASCs exposed to ANG-II [45]. MiR-26 might also inhibit inflammation by targeting *TLR4* [46]. However, inflammatory TNF-α, IL-6, and IL-1β inhibited the expression of miR-26a in ASCs, reducing their capacity to correct blood lipids [47]. The long noncoding RNA-small nucleolar RNA host gene 9 (lncRNA-SNHG9) in exosomes from ASCs reverted the inflammation and apoptosis of ECs by targeting the TNF receptor type 1-associated death domain protein mRNA [48].

ASCs inhibited the differentiation of peripheral blood mononuclear cells into IL-17-producing Th17 cells by inhibiting *RORγt*, the key transcription factor for Th17 cells. In contrast, they induced anti-inflammatory CD4^+^CD25^+^Foxp3^+^T regulatory (Treg) cells associated with an increase in IL-10 and TGF-β1 [49]. ASCs also blocked the differentiation of IFN-γ-producing inflammatory Th1 cells [50]. Of interest, rapamycin increased anti-inflammatory TGF-β and IL-10 [51]. MiR-10a-loaded exosomes increased TGF-β1 and decreased IFN-γ [52]. However, miR-10a was downregulated in obesity [53] (Figure 3).

## 5. Adipose-Derived Stem Cells Preserve Heart Function in Preclinical Models

Stromal cell-derived factor-1*α* (SDF-1*α*) and its receptor, C-X-C chemokine receptor type 4 (CXCR4), are critical for the recruitment, homing, and engraftment of transplanted ASCs into a myocardial infarction damage site. Activation of the SDF-1/CXCR4 axis by physical training potentiated stem cell therapy reduces vasoconstrictor and inflammatory responses [54]. SDF-1 released by ASCs increased the number of circulating endothelial progenitor cells (EPCs) and capillary density and reduced hind limb ischemia [55]. Periostin might increase these effects of ACSs, inducing integrin β1, PI3K/AKT, and eNOS [56] (Figure 4).

Ischemia-reperfusion (I/R) and hypoxia-reoxygenation (H/R) in a rat model triggered myocardial apoptosis through NF-κB, PUMA, and p53, downregulating BCL-2 and upregulating BAX and caspase 3. I/R- and H/R-induced heart damage was associated with fibrosis by inducing ETS-1, fibronectin, and collagen 3 [57]. Intramuscular injection of ASCs or exosomes from ASCs reduced this heart damage and fibrosis by reducing BAX and increasing BCL-2 and cyclin D1. They also inhibited PUMA, ETS-1, fibronectin, and collagen [58]. Of interest, rosuvastatin reinforced the action of ASCs [59]. ASCs in which prolyl hydroxylase domain protein 2 (PHD2), a cellular oxygen sensor, was silenced reduced the myocardial infarct size and prevented loss of function in mice by preventing cardiomyocyte cell death [60].

ASC-derived exosomes injected into the myocardium of I/R-treated mice significantly induced miR-221/222 and reduced levels of PUMA and ETS-1, which are associated with lower H_2_O_2_-induced apoptosis [61]. MiR-210 in exosomes from hypoxia-exposed ASCs inhibited cardiomyocyte apoptosis by blocking the expression of protein tyrosine phosphatase 1B and death-associated protein kinase 1 [62]. MiR-224-5p increased in exosomes derived from ASCs, downregulated TXNIP, and blocked apoptosis by sustaining the expression of *BCL-2* [63], while miR-301 inhibited the apoptosis signal-regulating kinase 1 (ASK1), decreasing ROS release [64]. Clathrin-mediated endocytosis of miR-214, enriched in the conditioned medium of ASCs, inhibited cardiomyocyte apoptosis [65]. Circ_0001747, enriched in exosomes from ASCs, elevated the messenger RNA and protein levels of the MCL1 apoptosis regulator, the BCL-2 family member (MCL1), by sequestering miR-199b-3p and attenuating H/R-induced injury [66].

ASCs restored the expression of integrin β_2_ that specifically blocks ICAM-1 and reduces macrophage accumulation in infarcted myocardium, increasing cell viability, proliferation, and migration [67]. In addition, ASC-derived exosomes mitigated MI-induced cardiac damage by promoting macrophage M2 polarization by restoring the expression of the sphingosine-1-phosphate receptor 1 (S1PR1) [68,69]. MiR-146a in ACS-derived exosomes inhibited hypoxia-induced TLR-4 and NF-κB and prevented inflammation-associated myocardial cell apoptosis and fibrosis [70]. MiR-181b also protected cardiomyocytes by suppressing inflammatory TLR4 and NF-κB, cell death signaling, and promoting *IGF1R* and *PI3K/AKT* signaling [71]. MiR-671 in ASC-derived exosomes targeted the transforming growth factor beta receptor 2 (TGFBR2) and suppressed the phosphorylation of SMAD2, enhancing cardiomyocyte viability and reducing myocardial fibrosis and inflammation [72]. X-inactive specific transcript (XIST) in exosomes from ASCs protected mice against myocardial pyroptosis and arterial fibrillation by reducing the activation of the NLRP3 inflammasome and the secretion of IL-1β and IL-18 in cardiomyocytes by targeting miR-214-3p [73]. 

ASCs attached to microparticles loaded with neuregulin (NRG) reduced infarct size, stimulated cardiomyocyte proliferation and the formation of arterioles and capillaries, and increased M2 macrophage polarization [74], ultimately promoting angiogenesis through miR-21 and CSF-1 [75]. Hypoxia improved the angiogenic capacity of ASCs by increasing VEGF-A and ANG-II [76]. In addition, exosomes from ASCs promoted angiogenesis by the delivery of miR-31, which targets *factor-inhibiting hypoxia-inducible factor-1* (*FIH1*) [77]. MiR-126 in ASC-derived exosomes prevented myocardial damage by preventing inflammation, apoptosis, fibrosis, and increasing angiogenesis [78]. Furthermore, miR-205 in exosomes from ASCs increased angiogenesis and improved cardiac function in MI-treated mice by inducing HIF-1α and VEGF [79] (Figure 4).

## 6. Adipose-Derived Stem Cells Are Less Efficient in the Human Clinical Setting Than in Preclinical Models

ASCs delivered to thirteen patients with ischemic heart failure and refractory angina who were not qualified for any form of direct revascularization did not improve LVEF and cardiac output at 12 months of follow-up [80]. In a Danish multi-center double-blinded placebo-controlled phase II study, direct intramyocardial injections of allogeneic ASCs were safe but did not improve myocardial function, structure, or clinical symptoms [81].

The phase II, randomized, double-blinded, placebo-controlled MyStromalCell trial included patients with chronic ischemic heart disease. ASCs did not improve myocardial perfusion, LVEF, myocardial mass, stroke volume, left ventricle end-diastolic volume, end-systolic volume, and the amount of scar tissue [82]. At 3 years follow-up, the bicycle exercise time and the exercise performance in watts were unchanged, but the performance measured in metabolic equivalents (METs) was slightly increased. In the same period, bicycle exercise time and exercise performance declined in the placebo group. Although angina was significantly reduced in the ASC group but not in the placebo group, there was no significant difference between the groups [83]. Furthermore, the intramyocardial delivery of ASCs that were stimulated by VEGF-A_165_ did not improve exercise ability compared to the placebo [84]. In yet another study including thirty-one patients (seventeen treated with autologous ASCs, fourteen with placebo), ASCs increased the maximal oxygen consumption (MVO2) but not LVEF, left ventricle end-diastolic volume, and end-systolic volume [85]. In the PRECISE Trial, a randomized, placebo-controlled, double-blind trial, transendocardial injections of ASCs preserved MRTs and MVO2s while they declined in the control group. The difference in the change in MVO2 from baseline to 6 and 18 months was significantly better in ASC-treated patients compared with the controls. The total left ventricular mass and wall motion score index improved in ASC-treated patients, and inducible ischemia was reduced after 18 months [86]. In the Therapeutic Angiogenesis by Cell Transplantation using ASCs (TACT-ADRC), the ASC cohort improved rest pain and 6 min walking distance. Circulating CD34^+^ and CD133^+^ progenitor cell markers increased. The ratio of VEGF-A_165_b (an anti-angiogenic isoform of VEGF) to total VEGF-A in plasma significantly decreased, as did the TNF-α in macrophages [87]. However, the goal of this study was to assess the effect of autologous ASCs on their ability to promote angiogenesis and suppress tissue inflammation more than their ability to improve heart function. 

In conclusion, in contrast to the preclinical models in which ASCs could preserve heart function in patients, they had limited effects. Therefore, we searched for reasons to explain these differences. Comparing the preclinical models and human settings, two main differences emerged. One, most often animals were young and exposed to ischemia for a brief period, whereas heart dysfunction in patients developed over a much longer time. Finally, the age-related metabolic risk factors that contribute to the development of human cardiovascular diseases are not reiterated in animal models. Therefore, in the last part, we looked at the effect of aging and metabolic reprogramming on the functionality of ASCs. 

## 7. Aging and Metabolic Reprogramming Decrease the Number and Function of Human ASCs

Aging is associated with increased oxidative stress and mitochondrial dysfunction-associated cellular damage, contributing to the decline in stemness. *Nicotinamide adenine dinucleotide* (*NAD+*), required for maintaining cellular homeostasis and stemness, decreases not only with age directly but also with age-related metabolic disorders. In addition, *NAD+* mitigates the differentiation of ASCs to mature adipocytes, associated with an increase in mitochondrial activity and ROS release [88]. The CD271-positive ASC subpopulation was reduced in the adipose tissues of diabetic patients, associated with decreased angiogenesis and the expression of the adipose stem cell marker SOX2 [89]. Metformin may prevent the differentiation of ASCs to adipocytes and slow down ASC proliferation, preventing these cells from proliferation exhaustion by enhancing the expression of stemness signature genes encoding *BMP7*, *dipeptidyl peptidase 4* (*DPP4 or T-cell activation antigen CD26*), *SOX2*, *OCT3/4* (or POU class 5 *homeobox* 1, *POU5F1*), *WNT2*, CD90, and *delta-like non-canonical Notch ligand 1 (DLK1)* [90]. Older ASCs had an inverse effect on T cell function by augmenting Th1 cells secreting IFN-γ and decreasing the percentage of anti-inflammatory Tregs [91]. Type 2 diabetes may further enhance this age-dependent effect [92]. Furthermore, age impaired the paracrine action of ASCs evidenced by reduced levels of SDF-1α, VEGF, and HGF [93]. In addition, TGF-β1 and proliferative rates of ASCs decreased with donor age [94]. However, another study revealed that a higher number of cell passages has a greater effect on the stemness of ASCs than donor age by itself; for example, by inducing the NF-κB signaling pathway closely related to harmful immune and inflammatory responses [95] (Figure 5).

MiR-34a is upregulated by high glucose [96]. MiR-34a induced senescence by targeting *SIRT1* [97,98,99]. In addition, miR-34a decreased the expression of various cell cycle regulators such as CDKs (-2, -4, -6), cyclins (-E, -D), and stem cell transcription factors KLF-4, OCT-4, SOX-2, and c-Myc. Thereby, it induced adipogenesis and lipid deposition. It increased inflammatory IL-6 and IL-8, further enhancing senescence [100]. In addition, miR-34a, inducing senescence, increased with an increasing number of cell passages [100,101]. MiR-486-5p was increased in obese subjects [102]. The blocking of its overexpression with exosomes from hypoxic ECs activated the AKT/MTOR/HIF-1α pathway, increasing the survival and engraftment of ASCs [103]. MiR-24-3p increased in obese and type 2 diabetes patients [104]. The downregulation of miR-24-3p in ASCs preserved ASC survival by restoring CDK4 expression and phosphorylated Rb protein levels [105]. Its inhibition also improved vascularization and reduced fibrosis after fat grafting [106]. However, ROS- and hypoxia-induced miR-210-3p in diabetic patients reversed the ANG-II-induced mitochondrial ROS accumulation and apoptosis in ASCs, decreased the cell death-inducing p53 target 1 and PKC/Raf-1/MAPK/NF-κB pathways, and increased the proliferation and migration of ASCs through the activation of PDGFR-β, AKT/ERK pathways [107,108,109,110]. MiR-17 was downregulated in diabetic patients [111]. Its downregulation induced oxidative stress and senescence in ASCs by the downregulation of stem cell markers c-Myc, OCT4, and Sca-1, and anti-oxidative HO-1 [112]. MiR-145 was also decreased in obese and diabetic patients, most probably due to a lack of TGF-β1 [113]. Restoring the expression of miR-145-5p in ASCs not only enhanced the expression of migration-associated protein FN1, the proliferation-associated proteins CCNA1 and CCND1, and the stem cell markers NANOG and OCT4 but also improved the functions of ECs and fibroblasts. In addition, miR-145 decreased the expression of senescence-associated protein p21 [114].

Higher LDL and lower HDL levels in patients with metabolic syndrome were associated with more ROS by the induction of oxidative NOX4 and NOX5 and more inflammatory molecules like MCP-1, C-C motif chemokine ligand 3 (CCL3 or MIP1α), and IL-8 [115]. In addition, obese- and particularly type 2 diabetes-derived ASCs released more inflammatory molecules due to the activation of NLRP3 inflammasome. Remarkably, the immunosuppressive activities of ASCs derived from obese and T2D subjects were reduced and associated with the less effective suppression of lymphocyte proliferation, activation of M2 macrophages, and TGF-β1 secretion than lean-derived ASCs. Treatment of human ASCs from lean subjects with IL-1β mimicked the dysfunctional immune behavior of obese and T2D human ASCs [116]. Exosomes isolated from ASCs enriched in miR-145 and mir-221 downregulated the pro-inflammatory markers IL-6 and NF-κB [44]. However, IFN-γ suppressed miR-221 [117]. In addition, exosomes enriched in miR-21 exerted an anti-inflammatory effect by blocking TLR4 and NF-kB signaling pathways [118]. However, miR-21 was downregulated by high glucose in diabetic patients [119]. In contrast, inflammatory IL-1β produced exosomes that transferred miR-146a to macrophages, protecting them against M1 polarization and reducing TNF-α and Il-6 by repressing NF-κB and AP-1 signaling [120]. Other studies suggested that hypoxia, very small-sized air particles, high glucose, TNF-α, and apolipoprotein E induce miR-146a [121,122,123,124,125,126,127,128,129].

Hyperglycemia, oxidative stress, altered immune reactions, and inflammation associated with type 2 diabetes limited the promotion of angiogenesis by ASCs [113,130]. Higher glycolysis may be responsible for their reduced angiogenic capacity because methylglyoxal (MGO), a highly reactive dicarbonyl primarily formed as a byproduct of glycolysis in chronic hyperglycemia and diabetes, inhibited VEGF and PDGF release. In addition, MGO induced the differentiation of ASCs to adipocytes, evidenced by the increased expression of PPARγ2 and increased Oil Red-O stainable lipids. 

Hypoxia induced let-7 in human ASC-derived extracellular vesicles via the let-7/argonaute 1/VEGF signaling pathway [131]. However, inflammation is associated with an increased expression of mitochondrial cytochrome c oxidase II (COX2), inducing the methylation of the promoter of let-7 and downregulating let-7 [132]. MiR-21 may increase angiogenesis and enhance its anti-oxidative effects against ROS damage by improving insulin sensitivity by targeting PTEN, inducing PI3K/AKT signaling [133,134]. In addition, miR-21 blocked TLR-4-mediated inflammation [135] (Figure 5).

## 8. Conclusions

ASCs proved to protect arteries and the heart in preclinical models in which cardiovascular risks associated with obesity, high glucose, low HDL, and high LDL were kept low. When in patients, these risk factors were not controlled; loss of endothelial integrity, oxidative stress, and inflammation may be associated with metabolic reprogramming of ASCs, leading to loss of functionality. Noncoding RNAs were shown to regulate ASC function, and differences in their expression may explain differences in outcomes in preclinical models and patients. Table 1 shows that a decrease in let-7, miR-17-92, miR-21, miR-145, and miR-221 led to the loss of their function with obesity, type 2 diabetes, oxidative stress, and inflammation. An increase in miR-34a, miR-486-5p, and mir-24-3p contributed to the loss of function, with a noteworthy increase in miR-34a with age. In contrast, miR-146a and miR-210 may protect stem cells. However, a systematic analysis of other noncoding RNAs in human adipose-derived stem cells is warranted. Overall, this review gives insight into modes to improve the functionality of human adipose-derived stem cells.

## Figures and Tables

**Figure 1 cells-12-02785-f001:**
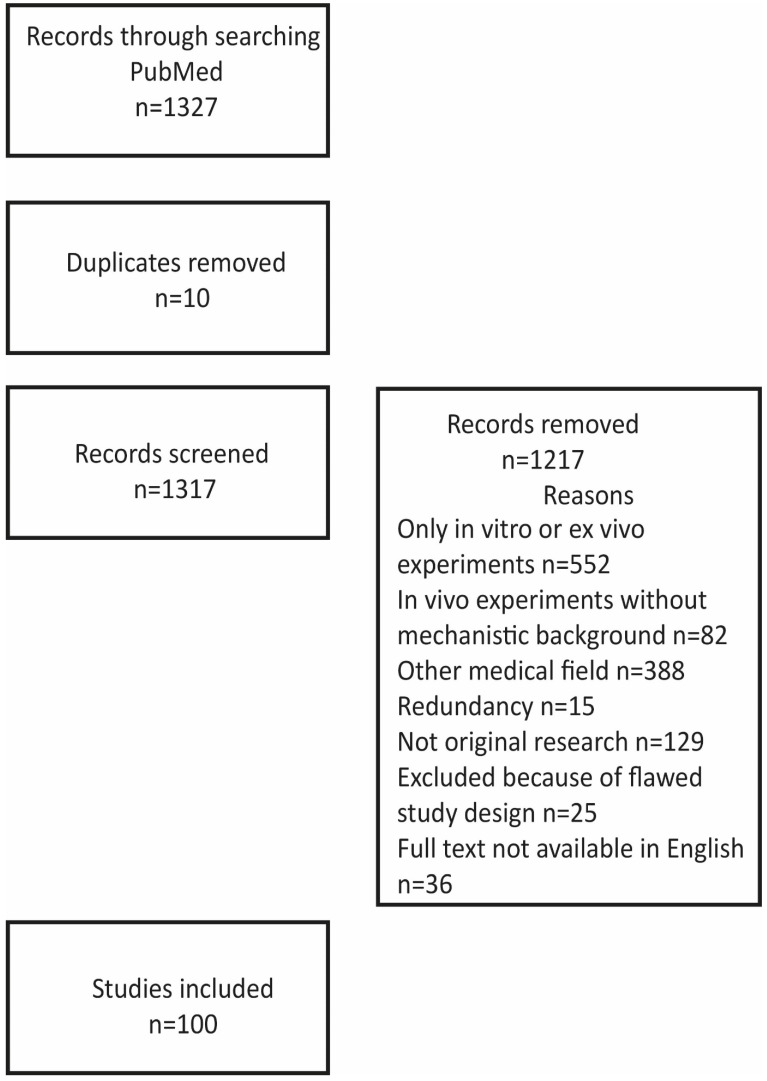
Strategy and outcome of the literature search.

**Figure 2 cells-12-02785-f002:**
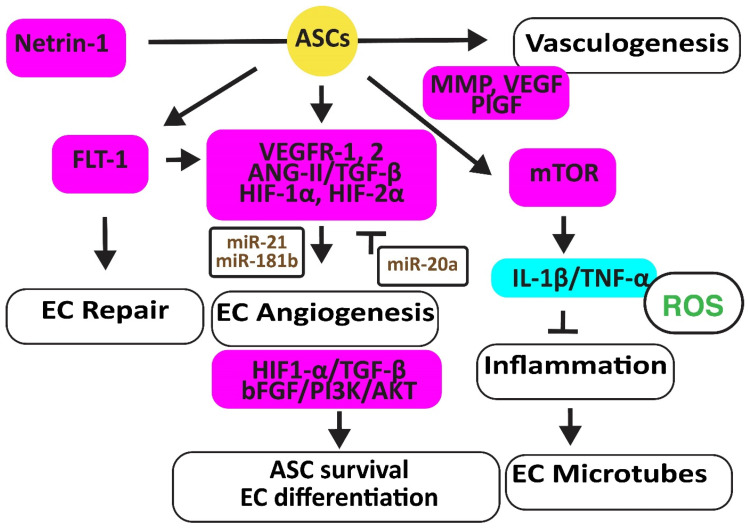
Adipose-derived stem cells improve EC function. The figure illustrates how adipose-derived stem cells preserve EC repair, angiogenesis, and microtube formation, and how functional ECs preserve the viability and differentiation ability of adipose-derived stem cell differentiation. Increased regulators are in purple boxes and decreased ones are in pale blue boxes. Upregulated noncoding RNAs are in brown. Decreased ROS is in green. Arrowheads reflect activation; hammerheads reflect inhibition.

**Figure 3 cells-12-02785-f003:**
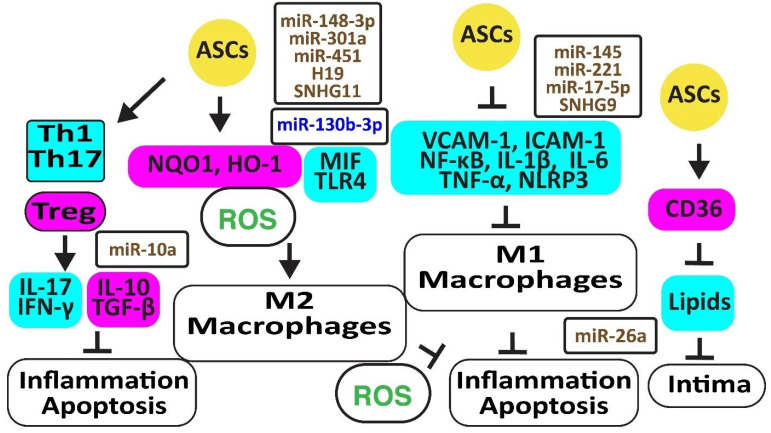
Adipose-derived stem cells prevent atherosclerosis. The figure illustrates how adipose-derived stem cells preserve protective M2 macrophages and Treg cells associated with decreased oxidative stress and inflammation. Increased regulators are in purple boxes and decreased ones are in pale blue boxes. Upregulated noncoding RNAs are in brown and downregulated ones are in dark blue. Decreased ROS is in green. Arrowheads reflect activation; hammerheads reflect inhibition.

**Figure 4 cells-12-02785-f004:**
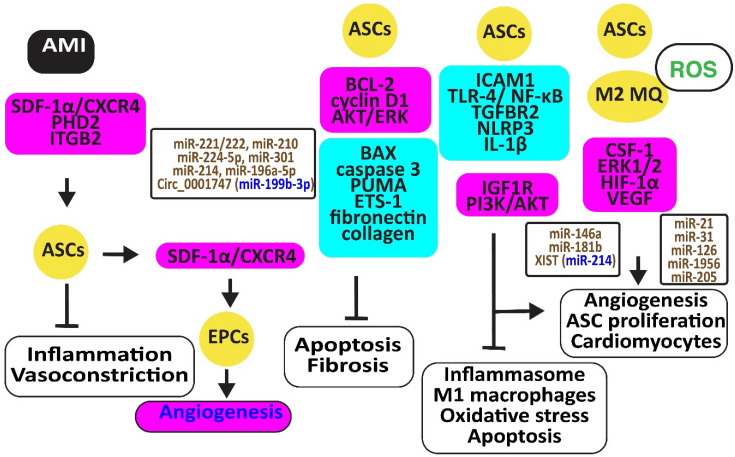
Adipose-derived stem cells preserve heart function. The figure illustrates how adipose-derived stem cells protect against AMI, injury caused by ischemia-reperfusion and hypoxia-reoxygenation by retaining M2 macrophages, reducing oxidative stress, inflammation, and apoptosis, preventing vasoconstriction, and restoring angiogenesis. Increased regulators are in purple boxes and decreased ones are in pale blue boxes. Upregulated noncoding RNAs are in brown and downregulated ones are in dark blue. Decreased ROS is in green. Arrowheads reflect activation; hammerheads reflect inhibition.

**Figure 5 cells-12-02785-f005:**
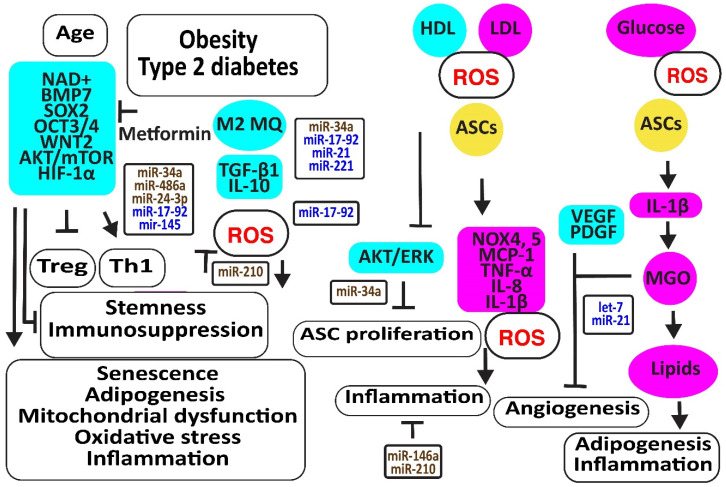
Aging and metabolic reprogramming decrease the number and function of human adipose-derived stem cells. The figure illustrates how aging, obesity, type 2 diabetes, and hyperlipidemia reduce stemness and induce senescence in ASCs associated with loss of function in association with oxidative stress, inflammation, apoptosis, and loss of angiogenesis. Increased regulators are in purple boxes and decreased ones are in pale blue boxes. Upregulated noncoding RNAs are in brown and downregulated ones are in dark blue. Increased ROS is in red. Arrowheads reflect activation; hammerheads reflect inhibition.

**Table 1 cells-12-02785-t001:** Overview of critical miRs in human adipose-derived stem cells and their regulation by aging and metabolic reprogramming.

MiR	Functions	Pathways	Observed Changes
**MiRs that protected but were downregulated**			
Let-7e	Induced angiogenesis	Induced VEGF	Downregulated by inflammatory markers and oxidative stress
miR-17-92	Protected against senescence Protected against mitochondrial dysfunction, oxidative stress, and inflammation	Upregulated stem cell markers c-Myc, OCT4, and SCA-1 Upregulated HO-1	Downregulated in type 2 diabetes patients
miR-21	Promotes angiogenesis and vascularization Inhibited inflammation	Upregulated HIF-1α, VEGF, and CSF-1 Blocked TLR-4	Downregulated in type 2 diabetes patients
miR-145	Preserved stem cell number Inhibited senescence Improved migration	Induced the stem cell markers NANOG and OCT4; induced the proliferation-associated proteins CCNA1 and CCND1 Induced FN1 Blocked p21	Decreased in obese and diabetic patients, most likely due to a lack of TGF-β1
miR-221	Inhibited inflammation	Inhibited IL-6 and NF-κB	Decreased by inflammatory IFN-γ
**MiRs that impaired functions and were upregulated**			
miR-34a	Induced senescence Blocked cell proliferation Blocked stemness and induced adipogenesis with lipid deposition Induced inflammation-enhancing senescence	Blocked SIRT1 Inhibited CDKs (-2, -4, -6) and cyclins (-E, -D) Inhibited KLF-4, OCT-4, SOX-2, and c-Myc Induced IL-6 and IL-8	Increased by aging and high glucose
miR-486-5p	Induced senescence	Inactivated mTORC1	Upregulated in obese subjects
miR-24-3p	Induced senescence	Blocked CDK4 expression and decreased phosphorylated Rb protein levels	Increased in obese and type 2 diabetes patients
** MiRs that protected and were upregulated **			
miR-210	Inhibited inflammation and senescence Protected against oxidative stress	Inhibited death-inducing p53 target 1 and blocked PKC/Raf-1/MAPK/NF-κB pathways Reversed the ANG-II-induced mitochondrial ROS	Increased in diabetic patients
miR-146a	Inhibited M1 macrophage polarization and inflammation	Repressed NF-κB and AP-1 signaling	Increased by hypoxia, high glucose, and inflammatory markers IL-1β and TNF-α

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
