# Peer review of "Aging and Metabolic Reprogramming of Adipose-Derived Stem Cells Affect Molecular Mechanisms Related to Cardiovascular Diseases"

_cells, 2023, doi:10.3390/cells12242785_

Round 1
Reviewer 1 Report
Comments and Suggestions for Authors
The manuscript by Holvoet entitled “Adipose-Derived Stem Cells: Molecular Mechanisms Behind the Discrepancy Between Preclinical Models and Clinical Treatment of Cardiovascular Diseases” reviews the impact of adipose-derived stem cells (ASCs) in preclinical models and clinical trials and the issues between their effectiveness in the preclinical models and lack of efficacy in clinical trials. The article summarizes published data derived from a systematic search of the PubMed database for English-language articles related to the function of ASCs. The review manuscript discusses insights into underlying molecular mechanisms for ASC-mediated protection against cardiovascular diseases and presents the rationale for failures in clinical research. The lack of efficacy is discussed mechanistically, including metabolic reprogramming, limited proliferation, loss of mitochondrial function, increased oxidative stress, and dysregulated immune response in ASCs. The manuscript devotes significant effort to focusing on noncoding RNAs that may be mechanistically related to these dysfunctions in ASCs.
While numerous review articles have summarized the impact of stem cells as therapeutics for cardiovascular disease, I think this one offers a thorough summary of the literature in the field up to the current time frame. It summarizes 100 published articles focused on this topic.
Suggestions for the author:
1. The introduction section describing ASCs is somewhat limited and perhaps should be more thoroughly presented for readers who may not be familiar with the cells.
2. The presentation of the impact on cardiovascular disease comes later in the manuscript. The rationale for presenting all mechanistic activities of ASCs is given first, but the rationale or description of how these mechanisms may impact cardiovascular diseases is lacking.
3. The figures encompassing critical pathways for each mechanism are beneficial and informative.
4. The section on human trials is only a small portion of the paper, but it is the main attraction of the paper.
5. The summary mechanisms that may result in differing human outcomes are an excellent addition.
Overall, the review is very good and should impact the field.
Author Response
I thank you for the overall positive evaluation of this review paper.
- The introduction section describing ASCs is somewhat limited and perhaps should be more thoroughly presented for readers who may not be familiar with the cells.
Response: The revised introduction contains more general background information on ASCs.
- The presentation of the impact on cardiovascular disease comes later in the manuscript. The rationale for presenting all mechanistic activities of ASCs is given first, but the rationale or description of how these mechanisms may impact cardiovascular diseases is lacking.
Response: The introduction contains more background information on the mechanisms of atherosclerosis and heart ischemia.
- The figures encompassing critical pathways for each mechanism are beneficial and informative.
Response: Figures 2-5 have been revised to even further increase their educational value.
- The section on human trials is only a small portion of the paper, but it is the main attraction of the paper.
Response: Unfortunately, only a few controlled trials have been published in the field of cardiovascular diseases.
- The summary mechanisms that may result in differing human outcomes are an excellent addition.
Response: Thank you. The new Figure 5 illustrates these mechanisms even better.
Reviewer 2 Report
Comments and Suggestions for Authors This is a well-prepared manuscript on a timely subject. The author intends to summarize developments that may propel the cell-free strategies, namely on exosomes secreted by adipose-derived stem cells (ADSCs) and their regulatory microRNA (miR) content, while identifying points that require further investigation. This is contextualized in a group of wide spread conditions that share at the basis critical cellular and molecular players such as the state of the endothelial compartment and the activity of the immune system. Disruption of each of these two axes imparts changes at the tissue and systemic levels as illustrated on the obesity, metabolic syndrome and cardiovascular diseases triad. The intertwined nature of these conditions has does been long appreciated and efforts multiply to contain the devastating social impact. The author focuses on the current R&D paradigm of exploring cell-free strategies, namely on exosomes secreted by adipose-derived stem cells (ADSCs) and on their regulatory microRNA (miR) content, toward improving the clinical management of a growing number of patients. To clarify the area and set the ground for the research ahead, departing from a couple of cellular and clinical features (Angiogenesis, Atherosclerosis, Myocardial Infarction, Cardiomyopathy,…), this work selects and revolves on a hundred studies recorded on the PubMed. Both, the information resources starting from the database and the criteria established for subsequent exploration and selection of the studies, are clearly exposed in a detailed enough manner. The targeted reports were published in English, presented some sort of mechanistic insight obtained from experiments conducted both in vitro and in vivo, bared statistic robustness and related to metabolic and cardiovascular diseases. Proliferation, survival and metabolic reprogramming of ADSCs are then summarized, with emphasis on the regulation exerted by miRs. A protective role of ADSCs on repairing the dysfunctional endothelial cell (EC), an early common denominator on the development of metabolic and cardiovascular malfunction, is discussed at the light of the “mesenchymal stem/stromal cell-derived” secretome. The author revisits evidence from animal-models to highlight the molecular players involved on anti-apoptotic, anti-inflammatory, angiogenic activities / overall processes that mitigate fibrosis and confer cardioprotection, before moving to the human trials data on ADSCs therapy for ischemic heart disease. Centring on the miRs action, a molecular basis for the understanding of the preclinical models and clinical contrasting results is then advanced. A closing remark is made on the need for elucidating further the regulation of ADSCs, particularly by a growing number of Noncoding RNA species. Overall, a fine and concise view is provided that can be of help for distinct kind of readers. MINOR: I find little appeal on the figures accompanying this manuscript; a lot better could be done although a colour-palette change would already improve, as the present primary colours used are too “heavy” and in some way less pleasant.Author Response
Thank you for finding our paper fine and concise.
MINOR Comment: I changed the color palette, revised the content of the Figures, and added a new Figure 5 to better explain the mechanisms behind the less-than-optimal behavior of human ASCs.